# The Mixture of *Saccharomyces cerevisiae* and *Clostridium butyricum* Could Promote Rumen Fermentation and Improve the Growth Performance of Goats in Hot Summer

**DOI:** 10.3390/metabo13010104

**Published:** 2023-01-09

**Authors:** Liyuan Cai, Min Li, Shuyi Zhou, Qingbiao Xu

**Affiliations:** 1College of Animal Science and Technology, Huazhong Agricultural University, Wuhan 430070, China; 2Institute of Dairy Science, College of Animal Science, Zhejiang University, Hangzhou 310058, China; 3Hainan Provincial Animal Husbandry Technology Promotion Station, Haikou 570100, China

**Keywords:** goats, hot summer, probiotics, rumen fermentation, growth performance

## Abstract

This study aimed to investigate the effects of multiple mixing ratio pairs of *Saccharomyces cerevisiae* (SC) and *Clostridium butyricum* (CB) on rumen fermentation and growth performance of goats in hot summer. Thirty goats were divided into five groups: 0.00% probiotics (control), 0.30% SC and 0.05% CB (P1), 0.30% SC and 0.10% CB (P2), 0.60% SC and 0.05% CB (P3), and 0.60% SC and 0.10% CB (P4) of the dry matter (DM) weight of the basal diet and were assigned to a 5 × 5 Latin square experimental design. The results showed the pH values, the activities of ruminal cellulolytic enzymes, and the concentrations of ammonia nitrogen, acetic acid, propionic acid, total volatile fatty acids, vitamins B1 and B2, and niacin were significantly increased (*p* < 0.05) by probiotics. Moreover, the DM intake, average daily gain, the digestibilities of DM, neutral detergent fiber, and acid detergent fiber were significantly increased (*p* < 0.05) in probiotic-supplemented groups. Additionally, among all probiotic supplementation groups, the P3 group had the most beneficial effect on rumen fermentation parameters and the growth performance of goats. These results suggested that the mixture of 0.60% *Saccharomyces cerevisiae* and 0.05% *Clostridium butyricum* of the DM concentration was beneficial to improve rumen fermentation and promote the growth of goats in hot summer.

## 1. Introduction

Goats are commonly kept in natural ventilation houses in the Jianghuai region of China [1]. Goats kept in this way are vulnerable to the adverse effects of summer temperatures, which mainly lead to the decrease in rumen fermentation function and ultimately lead to decreased growth performance [2]. 

Probiotics have been widely applied in livestock production because of their non-toxic side effects, no drug resistance, and no residue characteristics. *Saccharomyces cerevisiae* (a type of yeast) could improve the balance of beneficial gut microbiota and eliminate harmful pathogens, feed digestion, production performance, and health status [3,4]. Yeast is a critical probiotic group used in ruminant production [5]. It has outstanding advantages in stabilizing the rumen environment by providing nutrients to promote the growth of lactic acid bacteria for the stability of the rumen pH, increasing the population of cellulolytic bacteria [5,6]. *Clostridium butyricum* is a promising probiotic candidate, which could provide nutrients and antioxidants to balance the gut microbiota and improve immunity and growth performance in ruminants [7]. Compared to its wide application in monogastric animals, *Clostridium butyricum* is less used in ruminants [8,9]. It has been suggested that *Clostridium butyricum* could promote rumen fermentation and growth performance of calves and bred cattle [10]. 

Our previous study reported that supplementation with the mixture of *Saccharomyces cerevisiae* and *Clostridium butyricum* had a more beneficial effect on promoting rumen fermentation and improving the growth performance of goats in hot summer than the use of the single *Saccharomyces cerevisiae* or *Clostridium butyricum* [11]. However, only one level of combination of these two probiotics was used in our previous study. Therefore, this study aimed to investigate the effects of multiple mixing ratio pairs of *Saccharomyces cerevisiae* and *Clostridium butyricum* on rumen fermentation and growth performance of goats in hot summer. This study could provide a scientific and novel insight in using the mixture of *Saccharomyces cerevisiae* and *Clostridium butyricum* in hot summer to promote rumen fermentation and improve the growth performance of goats.

## 2. Materials and Methods

### 2.1. Goats and Management

This study was approved by the Animal Care and Use Committee of Huazhong Agricultural University (Approval code: HZAUGO-2015-008) and was carried out from May to October 2021. The study was conducted on 30 (5.0 ± 1.0 months) Macheng Black × Boer crossed goats, half male and half female, weighing 19 to 21 kg. These goats were kept in a naturally ventilated house with individual feeding pen (1.20 × 1.50 m) on an intensive farm in Hefei, China. Goats were fed on a basal diet in DM of 1.30 kg/day. These goats were fed the diet twice a day at 6:00–8:00 and 17:00–19:00). These goats had free access to water and were gavaged with water to ensure daily water intake attained 3.50 L per goat. The composition and nutritional levels of the basal diet are given in Table 1.

### 2.2. Probiotics Feeding Experiments

Thirty goats were divided into five groups and assigned to a 5 × 5 Latin square experimental design. Probiotics were supplemented with the diet, and these groups were as follows: 0.00% probiotics (control), 0.30% *Saccharomyces cerevisiae* and 0.05% *Clostridium butyricum* (P1), 0.30% *Saccharomyces cerevisiae* and 0.10% *Clostridium butyricum* (P2), 0.60% *Saccharomyces cerevisiae* and 0.05% *Clostridium butyricum* (P3), and 0.60% *Saccharomyces cerevisiae* and 0.10% *Clostridium butyricum* (P4) of the DM weight of the basal diet. The *Saccharomyces cerevisiae* for ruminants was provided by Angel Yeast Co., Ltd. (Yichang, China) and had a content of live cells at 2.0 × 10^10^ CFU/g. The *Clostridium butyricum* was provided by Huijia Biotechnology Co., Ltd. (Huzhou, China) and had a content of live cells at 1.0 × 10^8^ CFU/g. Five experimental cycles were included in this study, each cycle lasting fourteen days. In each cycle, five grams of chromium oxide was added to the diet as an external marker on days 11 to 13 to determine nutrient digestibility. To eliminate the influence of probiotics in last cycle, all the goats were fed a basal diet for 7 days between cycles.

### 2.3. Sample Collection and Measurement

Blood and rumen contents samples were collected on the last day (day 14) of each experimental cycle. The blood samples were collected in the morning after 24 h of fasting. Blood samples were centrifuged at 3000 rpm for 10 min to obtain serum. Rumen contents were collected in the morning after 4 h of feeding and were collected from all goats using a rubber stomach tube with a Jin Teng GM-0.33A vacuum pump (Tianjin, China). The rumen contents were filtered through four layers of gauze to obtain rumen fluid. Fecal samples were collected before the morning feedings on days 12 to 14 of each experimental cycle, and these samples from the same group were pooled. Samples were stored at −20 °C for further analysis. 

The rectal temperature, skin temperature, pulse, and respiratory rate were measured. In brief, rectal temperature and skin temperature were measured using a clinical thermometer (Kefu, Hunan, China) and an infrared thermometer (Omega, New York, NY, USA). Respiratory rate was measured by a stethoscope (Kefu, Hunan, China) for monitoring inhalations and exhalations around the thoracic area. Pulse was measured by a stethoscope placed ventrally. These measurements were taken at 8:00, 12:00, and 17:00 daily throughout each day of the fourteen-day experimental cycle. According to the report by Xia et al. [12], the activities or concentrations of serum superoxide dismutase (SOD), Glutathione peroxidase (GSH-Px), vitamin E, vitamin C, and malondialdehyde (MDA) were determined by commercial kits provided by obtained from the Nanjing Jiancheng Bioengineering Institute (Nanjing, China). The activities of alanine transaminase (ALT), aspartate transaminase (AST), creatine kinase (CK), the concentrations of urea nitrogen (BUN), glucose, triglyceride (TG), total protein (TP), cholesterol (CHOL), potassium ion (K^+^), sodium ion (Na^+^), and chloride ion (Cl^−^) in serum were determined by combining DiaSys Diagnostic System (Frankfurt, Germany) and an automatic biochemical analyzer (Hitachi 7100, Tokyo, Japan) following the instructions. The detection method of serum parameters of goats above is shown in Table 2. 

Rumen pH values were determined immediately when rumen contents were collected using a digital pH meter (Orion Technology Co., Ltd., Massachusetts, USA). The concentrations of ammonia nitrogen (NH_3_-N) and VFAs were determined as described by Maitisaiyidi et al. [13] and Yang et al. [14], respectively. The acetic acid to propionic acid ratio (A/P ratio) was calculated by the concentration of acetic divided by propionic acid concentration. According to Ciulu et al. [15], China National Standard GB/T 17819-1999, and GB/T 5009 197-2003, the concentrations of ruminal vitamins B1, B2, B6, and B12, and niacin were measured using a 2100-liquid chromatograph (Shimadzu, Kyoto, Japan). Separation was performed on an Agilent Eclipse XDB-C18 (250 × 4.6 mm, 5 μm) column (Agilent Technologies, Santa Clara, CA, USA). Each sample was prepared and injected in triplicate. The injection volume was 20 μL. The activities of avicelase, CMCaes, cellobiase, and xylanase were measured as described before [16]. 

Daily dry matter intake (DMI) and average daily gain (ADG) were measured as described by Cai et al. [11]. The contents of the DM, NDF, and ADF were measured in the feedstuff and feces as described by #930.15 in the AOAC and Goering and Van Soest [17,18], respectively. The digestibilities of DM, NDF, and ADF were calculated as follows: (nutrients content feedstuff–nutrients content in feces)/nutrients content in feedstuff × 100.

### 2.4. Statistical Analysis

Data were analyzed using R studio (v4.0.5) (GitHub Inc., San Francisco, CA, USA). The data of the rumen fermentation and growth performance parameters in the control group and each probiotic-supplemented group were analyzed using a one-way analysis of variance (ANOVA) followed by a post hoc Dunn test for multiple pairwise comparisons. *p* values of less than 0.05 were considered statistically significant.

## 3. Results

### 3.1. Probiotics Did Not Affect the Physiological Parameters of Goats in Hot Summer

The skin temperature, rectal temperature, pulse, and respiratory rate were not significantly different among the control and probiotics supplementation groups. The physiological parameters fed probiotics in hot summer are given in Table 3.

### 3.2. Probiotics Affected Blood Biochemistry

Supplementation with the mixture of *Saccharomyces cerevisiae* and *Clostridium butyricum*, the activities of GSH-Px and SOD, and the concentrations of TP were increased significantly (*p* < 0.05) compared to the control group. However, the concentrations of MDA were decreased significantly (*p* < 0.05) in each probiotic-supplemented group compared to the control group. However, the activity and the concentrations of these three parameters above were significantly increased compared to the control group (*p* < 0.05). The ascensional range of these parameters in P3 was more significant than that in other probiotic groups. Blood biochemical parameters of goats fed probiotics supplementation in hot summer are shown in Table 4.

### 3.3. Probiotics Improve Rumen Fermentation

The pH values, the concentrations of NH_3_-N, total volatile fatty acids (TVFA), acetic acid, propionic acid, butyric acid, and A/P ratio, and the activities of avicelase, CMCaes, cellobiase, and xylanase in the rumen of goats were significantly increased in each probiotics-supplemented group compared with that of the control group (*p* < 0.05). Moreover, the ascensional range of these parameters in the P3 group was greater than that in other probiotic groups. The parameters of rumen fermentation of goats fed probiotics in hot summer are shown in Table 5.

### 3.4. Probiotics Promoted Ruminal B Vitamins Production in Hot Summer

The concentrations of ruminal vitamins B1 and B2, and niacin were significantly increased in each probiotic-supplemented group compared with that of the control group (*p* < 0.05). These three B vitamins have higher concentrations (*p* < 0.05) in the P3 group than that in other probiotic groups. The concentrations of ruminal B vitamins of goats fed probiotics in hot summer are shown in Table 6**.**

### 3.5. Probiotics Improved Growth Performance of Goats in Hot Summer

The DMI, ADG, and digestibilities of DM, NDF, and ADF of goats were significantly increased in each probiotic-supplemented group compared with that of the control group (*p* < 0.05). Moreover, the ascensional range of these parameters in the P3 group was more significant than in other probiotic groups. The parameters of the growth performance of goats fed probiotics supplementation in hot summer are shown in Table 7.

## 4. Discussion

Since heat arises from rumen fermentation, ruminants exhibit less tolerance to hot environments [19]. Rumen fermentation and the growth performance of goats could be adversely affected in hot summer (Figure 1) [2,20]. Moreover, heat stress can cause the disorder of redox balance of the body, the occurrence of oxidative stress, and damage to cells and tissues, thus affecting the growth and development of the body and health status [21]. Dietary supplementation with *Saccharomyces cerevisiae* and *Clostridium butyricum* was one of the most effective ways to alleviate the adverse effects of heat stress in goat production [11]. In this study, *Saccharomyces cerevisiae* and *Clostridium butyricum* were paired with different concentration gradients to study the promoting effect of their mixture on rumen fermentation and growth performance of goats in hot summer. This study is a continuation of our previous study showing that the two probiotics worked better in combination than individual. Therefore, it is necessary to screen out the optimal mixed dose through this study.

Rumen pH value is an important parameter closely related to the rumen environment, which can reflect the fermentation of diets in the rumen [22]. In this study, supplementation with the mixture of *Saccharomyces cerevisiae* and *Clostridium butyricum* could enhance the rumen pH of goats in hot summer. This result was consistent with previous studies on cows and goats with live yeast or *Saccharomyces cerevisiae* and *Clostridium butyricum* supplementation [11,23,24,25]. These results suggested that the mixture of these two probiotics could effectively promote the pH in hot summer. This may be because probiotics can promote the abundance of rumen bacteria and protozoa, which could consume lactic acid in the rumen [26,27]. In this study, supplementation with the mixture of *Saccharomyces cerevisiae* and *Clostridium butyricum* could increase the NH_3_-N concentration in the rumen, which was similar to previous studies [11]. The effects of these two probiotics on increased ruminal NH_3_-N could attribute to the rumen microbiota promoted by probiotics supplementation to degrade and hydrolyze protein [28]. However, previous studies also showed that yeast had no effect or could decrease the NH_3_-N concentration in the rumen [24,28,29,30]. The variety in the results can be attributed to the management system, species, and physiological state of the ruminants, probiotic species, supplemented levels of probiotics, the composition of the diets, and environmental conditions in different studies [11]. In this study, supplementation with the mixture of *Saccharomyces cerevisiae* and *Clostridium butyricum* improved the concentrations of VFAs in the rumen of goats in hot summer, similar to previous reports with consistent results [11,20]. However, previous studies also reported that these two probiotics do not affect ruminal VFAs concentrations [27,31,32]. Ruminal VFAs originate from the fermentation of feed fiber, and probiotics are beneficial for the growth and reproduction of ruminal cellulolytic bacteria, which enhances VFA production in the rumen [29,33]. Classical ruminant nutrition theory holds that the rumen microbiota could produce sufficient amounts of vitamin B to meet the needs of their ruminant host [34]. The function of B vitamins is to act as precursors for cofactors or enzyme cofactors [35]. Studies in recent years have shown that in certain physiological or rearing conditions, the production of ruminal B vitamins could not meet the requirements of ruminants [36]. Despite the deficiency of B vitamins, many studies have obtained expected results by supplementation with exogenous B vitamins. Previous studies reported that with dietary supplementation with vitamin B1 to dairy cows with ruminal acidosis, the rumen pH was increased, and the acidosis symptoms were eased [37,38]. Dietary supplements with niacin significantly decreased somatic cell numbers in heat-stressed cows [39]. Few studies have been conducted to supply B vitamins by feeding probiotics to ruminants. In this study, dietary supplementation with the mixture of *Saccharomyces cerevisiae* and *Clostridium butyricum* could significantly improve the yield of vitamin B1 and B2 and niacin in the rumen of goats in hot summer. The reasons for this effect of the mixture of these two probiotics can be summarized as follows: (1) yeast cells contain carbohydrates, such as glucose, furan mannose, and chitin, which can be used as fermentation substrates for ruminal microorganisms [40,41]; (2) yeast contains B vitamins; (3) yeast also directly provides the B vitamins for rumen microbiota and ruminants [42]; (4) *Clostridium butyricum* could provide product B vitamins as its metabolite [43,44]. In this study, the P3 group showed the most significant improvement in rumen fermentation. Therefore, the combination levels of 0.60% *Saccharomyces cerevisiae* and 0.05% *Clostridium butyricum* of dry matter are optimal supplemental levels to improve rumen fermentation in hot summer among all the probiotics groups.

In this study, dietary supplementation with the mixture of *Saccharomyces cerevisiae* and *Clostridium butyricum* promoted the digestibilities of DM, NDF, and ADF of goats in hot summer. Our previous studies also reported a similar result with supplementation with *Saccharomyces cerevisiae* and *Clostridium butyricum* separately of heat-stressed goats [11,20]. These promoted effects may be attributed to these probiotics that could consume O_2_ in the rumen and provide nutrients for ruminal cellulolytic bacteria [29,31]. Moreover, in this study, supplementation with the mixture of *Saccharomyces cerevisiae* and *Clostridium butyricum* enhanced the activities of avicelase, CMCase, cellobiase, and xylanase and could also be the plausible explanation for the increased digestibilities of DM, NDF, and ADF [11]. In this study, the combination levels of 0.60% *Saccharomyces cerevisiae* and 0.05% *Clostridium butyricum* of DM are the optimal supplemental levels to promote rumen fermentation and improve the growth performance of goats in hot summer. 

## 5. Conclusions

In hot summer, dietary supplementation with this mixture of *Saccharomyces cerevisiae* and *Clostridium butyricum* promoted the activities of serum GSH-Px and SOD, and the TP concentration while reducing serum MDA concentration. Moreover, supplementation with this mixture could effectively promote rumen fermentation by increasing pH, the concentrations of NH_3_-N, VFAs, vitamin B, and cellulolytic enzyme activities in the rumen of goats. Additionally, this mixture could effectively improve growth performance by improving the digestibilities of DM, NDF, and ADF in goats. To promote rumen fermentation and enhance the growth performance of goats in hot summer, the combination levels of 0.60% *Saccharomyces cerevisiae* and 0.05% *Clostridium butyricum* of the DM concentration in the basal diet are optimal supplemental levels. 

## Figures and Tables

**Figure 1 metabolites-13-00104-f001:**
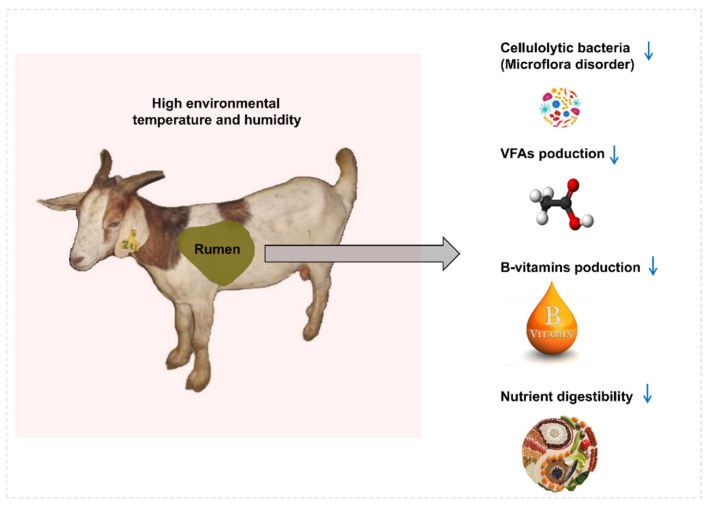
The adverse effects of hot summer on rumen fermentation and growth performance of goats. The blue arrow indicated a decrease.

**Table 1 metabolites-13-00104-t001:** Composition and nutritional levels (g/kg) of the basal diet.

Ingredient	Content	Nutrition Level	Amount
Alfalfa	564	Dry matter	955
Ground corn	261	Organic matter	852
Soybean meal	80	Crude protein	177
Wheat barn	78	Neutral detergent fiber	436
Ca_2_HPO_4_	7	Acid detergent fiber	262
Premix *	10	Ca	5.9
		P	3.2

* Premix contained per kg: 20.70 g Mg, 0.50 g Fe, 1 g Mn, 2 g Zn, 43 mg Se, 47 mg I, 54 mg, Co, 90,000 IU vitamin A, 17,000 IU vitamin D, and 1750 IU vitamin E.

**Table 2 metabolites-13-00104-t002:** The detection methods of serum parameters of goats.

Biochemical Index	Methods
SOD, GSH-Px, vitamin C and E, and MDA	Spectrophotometry
ALT, AST	Ultraviolet continuous monitoring
BUN	Glutamate dehydrogenase
CK	IFCC continuous monitoring
Glucose	Glucose oxidase
TP	Biuret method
TG, CHOL	Enzyme reagent
K^+^, Na^+^, Cl^−^	Mercury thiocyanate endpoint method

**Table 3 metabolites-13-00104-t003:** Physiological parameters of goats fed probiotics in hot summer.

Physiological Parameters	Control	P1	P2	P3	P4	SEM
Skin temperature (°C)	36.70	36.42	36.33	36.51	36.62	0.47
Rectal temperature (°C)	39.41	39.52	39.43	39.32	39.46	0.33
Pulse (beats/min)	84.73	84.75	83.87	85.57	84.23	2.43
Respiratory rate (breaths/min)	34.75	33.23	35.24	33.89	33.88	1.11

0.00% probiotics (control), 0.30% SC and 0.05% CB (P1), 0.30% SC and 0.10% CB (P2), 0.60% SC and 0.05% CB (P3), and 0.60% SC and 0.10% CB (P4) of the DM weight of the basal diet.

**Table 4 metabolites-13-00104-t004:** Blood biochemical parameters of goats fed probiotics in hot summer.

Parameters	Control	P1	P2	P3	P4	SEM
ALT (IU L^−1^)	16.26	16.11	16.17	16.19	16.22	0.33
AST (IU L^−1^)	81.37	75.71	71.65	74.73	72.17	3.21
CK (IU L^−1^)	204.4	184.5	187.2	186.6	185.5	17.44
Glucose (mmol L^−1^)	3.46	3.27	3.24	3.42	3.37	0.45
TP (g L^−1^)	58.17 ^a^	71.79 ^b^	70.83 ^b^	76.14 ^b^	70.78 ^b^	10.33
TG (mmol L^−1^)	0.38	0.41	0.40	0.43	0.39	0.06
BUN (mmol L^−1^)	8.87	8.74	8.34	7.87	8.46	0.55
CHOL (mmol L^−1^)	1.77	1.71	1.89	1.91	1.80	0.31
K^+^ (mmol L^−1^)	4.43	4.52	4.36	4.67	4.56	0.23
Na^+^ (mmol L^−1^)	142.1	144.4	144.8	146.8	144.6	17.34
Cl^−^ (mmol L^−1^)	102.5	102.7	102.7	103.4	102.0	7.21
SOD (IU mL^−1^)	170.8 ^a^	267.1 ^b^	264.7 ^b^	290.4 ^b^	276.8 ^b^	21.33
GSH-Px (µmol L^−1^)	249.7 ^a^	448.3 ^b^	469.8 ^b^	511.3 ^b^	469.3 ^b^	27.11
Vitamin C (µg mL^−1^)	20.7	38.8	38.97	35.90	37.84	2.14
Vitamin E (µg mL^−1^)	16.73	22.34	23.08	23.83	20.98	1.77
MDA (nmol mL^−1^)	8.41 ^a^	5.71 ^b^	5.82 ^b^	5.24 ^b^	5.21 ^b^	0.72

0.00% probiotics (control), 0.30% SC and 0.05% CB (P1), 0.30% SC and 0.10% CB (P2), 0.60% SC and 0.05% CB (P3), and 0.60% SC and 0.10% CB (P4) of the DM weight of the basal diet. Different letter superscripts indicate significant differences (*p* < 0.05).

**Table 5 metabolites-13-00104-t005:** The parameters of rumen fermentation of goats fed probiotics in hot summer.

Parameters	Control	P1	P2	P3	P4	SEM
pH	6.60 ^a^	6.81 ^b^	6.85 ^b^	6.88 ^b^	6.80 ^b^	0.05
NH_3_-N (mg 100 mL^−1^)	5.49 ^a^	8.34 ^b^	8.21 ^b^	9.43 ^b^	8.02 ^b^	0.41
**VFAs** (mmol L^−1^)						
TVFA	30.16 ^a^	50.56 ^b^	52.54 ^b^	57.05 ^b^	50.50 ^b^	3.22
Acetic acid	12.92 ^a^	25.14 ^b^	27.02 ^b^	28.82 ^b^	24.12 ^b^	1.54
Propionic acid	10.38 ^a^	15.41 ^b^	15.65 ^b^	17.02 ^b^	16.33 ^b^	1.05
Butyric acid	7.23 ^a^	10.11 ^b^	9.87 ^b^	11.21 ^b^	10.05 ^b^	0.23
A/P ratio	1.29 ^a^	1.64 ^b^	1.73 ^b^	1.69 ^b^	1.48 ^b^	0.04
Avicelase	1.31 ^a^	2.54 ^b^	2.44 ^b^	2.92 ^b^	2.34 ^b^	0.47
CMCaes	1.36 ^a^	2.69 ^b^	2.47 ^b^	3.21 ^b^	2.39 ^b^	0.12
Cellobiase	2.44 ^a^	4.77 ^b^	4.83 ^b^	5.24 ^b^	4.37 ^b^	0.43
Xylanase	4.54 ^a^	6.51 ^b^	6.42 ^b^	7.41 ^b^	6.21 ^b^	0.51

0.00% probiotics (control), 0.30% SC and 0.05% CB (P1), 0.30% SC and 0.10% CB (P2), 0.60% SC and 0.05% CB (P3), and 0.60% SC and 0.10% CB (P4) of the DM weight of the basal diet. Different letter superscripts indicate significant differences (*p* < 0.05).

**Table 6 metabolites-13-00104-t006:** The concentrations of ruminal B vitamins of goats fed probiotics supplementation in hot summer.

Vitamins	Control	P1	P2	P3	P4	SEM
(μg/mL) B1	0.14 ^a^	0.28 ^b^	0.24 ^b^	0.38 ^c^	0.24 ^b^	0.04
B2	0.19 ^a^	0.39 ^b^	0.69 ^b^	0.92 ^c^	0.52 ^b^	0.10
B6	1.52	1.72	1.50	1.85	1.23	0.13
B12	2.79	2.66	2.57	2.77	2.74	0.34
Niacin	8.12 ^a^	35.00 ^b^	33.51 ^b^	45.22 ^c^	39.43 ^b^	3.16

0.00% probiotics (control), 0.30% SC and 0.05% CB (P1), 0.30% SC and 0.10% CB (P2), 0.60% SC and 0.05% CB (P3), and 0.60% SC and 0.10% CB (P4) of the DM weight of the basal diet. Different letter superscripts indicate significant differences (*p* < 0.05).

**Table 7 metabolites-13-00104-t007:** The parameters of growth performance of goats fed probiotics in hot summer.

Parameters	Control	P1	P2	P3	P4	SEM
DMI (kg)	0.80 ^a^	1.16 ^b^	1.20 ^b^	1.25 ^b^	1.21 ^b^	0.06
ADG (kg)	0.08 ^a^	0.11 ^b^	0.13 ^b^	0.18 ^b^	0.14 ^b^	0.04
**Digestibility**						
DM (%)	36.67 ^a^	52.34 ^b^	50.21 ^b^	56.32 ^b^	52.12 ^b^	2.54
NDF (%)	30.17 ^a^	44.35 ^b^	45.43 ^b^	47.87 ^b^	43.21 ^b^	3.02
ADF (%)	30.54 ^a^	43.21 ^b^	43.22 ^b^	45.21 ^b^	42.22 ^b^	2.37

0.00% probiotics (control), 0.30% SC and 0.05% CB (P1), 0.30% SC and 0.10% CB (P2), 0.60% SC and 0.05% CB (P3), and 0.60% SC and 0.10% CB (P4) of the DM weight of the basal diet. Different letter superscripts indicate significant differences (*p* < 0.05).

## Data Availability

Not applicable.

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
