# Peer review of "The Mixture of Saccharomyces cerevisiae and Clostridium butyricum Could Promote Rumen Fermentation and Improve the Growth Performance of Goats in Hot Summer"

_metabolites, 2023, doi:10.3390/metabo13010104_

Round 1

Reviewer 1 Report

Comments to metabolites-2089468

This manuscript showed that the mixture of Saccharomyces cerevisiae and Clostridium butyricum could promote rumen fermentation and improve the growth performance of goats in hot summer, suggesting a proper proportion of mixed probiotics were beneficial to protect goats from heat stress in hot summer. This study provide adequate data and results to support the conclusion, which comes up with a good understanding of the effect of Saccharomyces cerevisiae and Clostridium butyricum. It added some novelty to the current study in the application of probiotics, which has a great potential to prevent heat stress. However, some details should be revised to make the it more clear.

    What's the standard of the concentrations of the probiotics treatment chosen in this study?

    Detailed review report: Some detail issues and language need to be addressed and optimized as follows:

Line 17-18: Change symbol ";" to ",". Line 19: Change "vitamins B1, B2, and niacin were significantly increased" to "vitamins B1 and B2, and niacin were significantly increased by probiotics ".

Line 20: Delete the first "and".

Line 23: Change "growth performance" to "growth performance of goats".

Line 24: What's the proper proportion of mixed probiotics? Change it to exact name.

Line 33-35: Change this sentence to "Our previous studies reported that when goats are kept in a high-temperature environment for a period of time, their ruminal pH, concentrations of ammonia-N, and volatile fatty acids (VFAs) were decreased significantly"

Line 37: Change "decreased" to "were decreased".

Line 41: Change "improving feed digestion, production performance, and health status [3. 4]" to "to improve feed digestion, production performance, and health status [3, 4]".

Line 45: Change " bacteria. Therefore improve..." to " bacteria, resulting in improving...".

Line 46-47: Rewrite this sentence as: "Compared to its wide application in monogastric animals, Clostridium butyricum is less used in ruminants [7-9]."

Line 64: Delete "last year".

Line 73: Delete one blank space in this sentence.

Line 98: Change "According to Xia et al." to "According to the report by Xia et al."

Line 99: Delete "concentrations".

Line 101: Change "The alanine..." to " The concentration of alanine...".

Line 102: Change "CHOL (cholesterol)" to "cholesterol (CHOL)".

Line 106: Change "is" to "was".

Line 110 and 150: Change "NH3-N" to "NH3-N". Delete "(VFAs)".

Line 112: Change "17819-1999 ," to "17819-1999," (Delete one blank space in this sentence).

Line 117: Change "Daily" to "Dry".

Line 118-119: Change this sentence to "The contents of the dry matter (DM), neutral detergent fiber (NDF), and acid detergent fiber (ADF) were measured in the feedstuff and feces as described by...".

Line 121-122: Change this sentence to "...(nutrients content feedstuff-nutrients content in feces)/nutrients content in feedstuff × 100".

Line 127: Delete "were performed".

Line 130: Change "do" to "did".

Line 131: Change "Rectal" to "rectal".

Line 132: Delete "(P1-P4 groups".

Line 138: Delete "activity".

Line 139: Add "were" before "increased".

Line 140-143: There is a blank space before "(". And all the value "p" in the whole manuscript should be italic.

Line 146 Change "...are given in Table 4" to "...were shown in Table 4.".

Line 154: Add "that" after "than".

Line 155, 163, 172 and other places: Change "are shown" to "were shown".

Line 167: Change "improve" to "improved".

Line 176: Change " " to " ".

Line 182: Delete "Dietary supplementation with".

Line 187: Change "because our previous showed" to "showing".

Line 188: Change "separately" to "individual".

Line 192: Change "...hot summer. Previous studies reported consistent results" to "...hot summer, as previous reports with consistent results".

Line 205: Add "was" before "increased" and "eased".

Line 210: Change "improve" to " improved".

Line 214: Change "which" to "yeast".

Line 215: Change "3)" to " 4)".

Line 216: Add "use" before "B-vitamins".

Line 223: Change "same result" to "similar results".

Line 225: Change "These promote effects could contribute to..." to "These promoted effects may be contributed from...".

Line 227: Change "bacteria. Therefore enhance..." to "bacteria, therefore enhancing...".

In the discussion section, a reference should be cited. Xu X, Wei Y, Hua HW, Jing XQ, Zhu HL, Xiao K, Zhao JC, Liu YL*. 2022. Polyphenols sourced from Ilex latifolia Thunb. relieve intestinal injury via modulating ferroptosis in weanling piglets under oxidative stress. Antioxidants. 11:966.

Author Response

Response to Reviewer 1 Comments

Dear reviewer, 

Thanks for your thoughtful comments and suggestions. As shown in the manuscript, we have made careful modifications to the manuscript. We have made thorough revisions to improve the English as needed. Below you will find our general reply and point-by-point responses to the reviewers’ comments/ questions.

Point 1: What's the standard of the concentrations of the probiotics treatment chosen in this study?

Response 1: In our previous studies, we separately conducted probiotics- supplemented experiments with multiple levels and found that dietary supplementation with 0.30% and 0.60% Saccharomyces cerevisiae and 0.05%  0.10% Clostridium butyricum, respectively could effectively promote rumen fermentation and improve growth performance of heat-stressed goats. Therefore, the effects of their mixtures have been studied in this study.

Point 2: Line 17-18: Change symbol “;” to “,”.

Response 2: “;” has been replaced to “,”on lines 17-18.

Point 3: Line 19: Change “vitamins B1, B2, and niacin were significantly increased” to “vitamins B1 and B2, and niacin were significantly increased by probiotics”.

Response 3: “vitamins B1, B2, and niacin were significantly increased” has been replaced with “vitamins B1 and B2, and niacin were significantly increased by probiotics” in line 19.

Point 4: Line 20: Delete the first “and”.

Response 4: the first “and” has been deleted at line 20.

Point 5: Line 23: Change “growth performance” to “growth performance of goats”.

Response 5: “growth performance” has been changed to “growth performance of goats” in line 24.

Point 6: Line 24: What's the proper proportion of mixed probiotics? Change it to exact name.

Response 6: “the proper proportion of mixed probiotics” has been replaced by “the mixture of 0.60% Saccharomyces cerevisiae and 0.05% Clostridium butyricum of the dry matter concentration” in lines 24-26.

Point 7: Line 33-35: Change this sentence to “Our previous studies reported that when goats are kept in a high-temperature environment for a period of time, their ruminal pH, concentrations of ammonia-N, and volatile fatty acids (VFAs) were decreased significantly”

Response 7: The original sentence has been replaced according to your suggestion in lines 35-38.

Point 8: Line 37: Change “decreased” to “were decreased”.

Response 8: “decreased" has been changed to "were decreased” at line 38.

Point 9: Line 41: Change “improving feed digestion, production performance, and health status [3. 4]” to “to improve feed digestion, production performance, and health status [3, 4]”.

Response 9: “improving feed digestion, production performance, and health status [3. 4]” has been replaced to improve feed digestion, production performance, and health status [3, 4]” at line 44.

Point 10: Line 45: Change “ bacteria. Therefore improve...” to “bacteria, resulting in improving...”.

Response 10: “bacteria. Therefore improve...” has been replaced to “bacteria, resulting in improving...” at line 47.

Point 12: Line 64: Delete “last year”.

Response 12: “last year” has been replaced by 2021 at line 66.

Point 13: Line 73: Delete one blank space in this sentence.

Response 13 : The blank space has been deleted at line 75.

Point 14: Line 98: Change “According to Xia et al.” to “According to the report by Xia et al.”

Response 14 : According to your suggestion, the referenceite has been replaced by “According to the report by Xia et al.” at line 107.

Point 15: Line 99: Delete “concentrations”.

Response 15: “concentrations” have been deleted at line 109.

Point 16: Line 101: Change “The alanine...” to “ The concentration of alanine...”.

Response 16: The sentence has been rewritten from lines 111 to 115.

Point 17: Line 102: Change “CHOL (cholesterol)" to “cholesterol (CHOL)”.

Response 17: “CHOL (cholesterol)” has been changed to “cholesterol (CHOL)” in line 113.

Point 18: Line 106: Change “is” to “was”.

Response 18: “is” was replaced with “was” at line 116.

Point 19: Line 110 and 150: Change “NH3-N” to “NH3-N”. Delete “(VFAs)”.

Response 19 : According to your suggestion, “NH3-N” has been changed to “NH3-N”, “volatile fatty acids” has been deleted, total volatile fatty acids have been added, and “p < 0.05” was changed to “p < 0.05” throughout the manuscript.

Point 20: Line 112: Change “17819-1999 ,” to “17819-1999,” (Delete one blank space in this sentence).

Response 20: The blank has been deleted at line 124.

Point 21: Line 117: Change “Daily” to “Dry”.

Response 21: I’m sorry! We believed that average daily gain (ADG) is an accurate description, so no modification is made.

Point 22: Line 118-119: Change this sentence to “The contents of the dry matter (DM), neutral detergent fiber (NDF), and acid detergent fiber (ADF) were measured in the feedstuff and feces as described by...”.

Response 22: According to your suggestion, this sentence has been changed in lines 131-132.

Point 23: Line 121-122: Change this sentence to “...(nutrients content feedstuff-nutrients content in feces)/nutrients content in feedstuff × 100”.

Response 23: According to your suggestion, this sentence has been changed in lines 134-135.

Point 24:Line 127: Delete “were performed”.

Response 24:“were performed” has been deleted at line 141.

Point 25: Line 130: Change “do" to “did”.

Response 25: “do” has been changed to “did” at line 143.

Point 26: Line 131: Change “Rectal” to “rectal”.

Response 26: “Rectal” has been changed to “rectal” in line 144.

Point 27: Line 132: Delete “(P1-P4 groups”.

Response 27: “(P1-P4 groups” has been deleted at line 145.

Point 28:Line 138: Delete “activity”.

Response 28: “activity” has been deleted at line 150.

Point 29: Line 139: Add “were” before “increased”.

Response 29: “were” has been added at line 150.

Point 30: Line 140-143: There is a blank space before “(”. And all the value “p” in the whole manuscript should be italic.

Response 30: According to your suggest, these misrepresentations have been corrected in the full text.

Point 31: Line 146 Change “...are given in Table 4” to “...were shown in Table 4.”.

Response 31: “..are given in Table 4? has been replaced with”..” were shown in Table 4.” at line 158.

Point 32: Line 154: Add “that” after “than”.

Response 32: “that” has been added at line 168.

Point 33: Line 155, 163, 172 and other places: Change “are shown” to “were shown”.

Response 33: “are shown” has been changed to “were shown” in lines 169, 179, and 189.

Point 34: Line 167: Change “improve” to “improved”.

Response 34: “improve” has been changed to “improved” at line 184.

Point 35: Line 176: Change “ ”to “ ”.

Response 35: I don't know what you mean.

Point 36: Line 182: Delete “Dietary supplementation with”.

Response 36: “Dietary supplementation with” has been deleted at line 199.

Point 37: Line 187: Change “because our previous showed” to “showing”.

Response 37: “because our previous showed” has been replaced with “showing” at 207.

Point 38: Line 188: Change “separately” to “individual”.

Response 38: “separately” has been changed to “individual” in line 208.

Point 39: Line 192: Change “...hot summer. Previous studies reported consistent

results” to “...hot summer, as previous reports with consistent results”.

Response 39: As your suggestion, this sentence has been replaced at line 312.

Point 40: Line 205: Add “was” before “increased” and “eased”.

Response 40: “was” were added at line 225 and 226.

Point 41: Line 210: Change “improve" to “ improved”.

Response 41: “improve” had been replaced with “ improved” at line 230.

Point 42: Line 214: Change “which” to “yeast”.

Response 42: “which” has been replaced by “yeast” at line 234.

Point 43: Line 215: Change “3)” to “4)”.

Response 43: “3)” has been changed to “ 4)” in line 235.

Point 44: Line 216: Add “use" before “B-vitamins”.

Response 44: “product” has been added before “B-vitamins” at line 236.

Point 45: Line 223: Change “same result” to “similar results”.

Response 45: “same result” has been replaced by “similar results” at line 243.

Point 46: Line 225: Change “These promote effects could contribute to....” to “These

promoted effects may be contributed from...”.

Response 46: The original sentence has been replaced by “These promoted effects may be contributed from...”. in line 245-246.

Point 47:  Line 227: Change “bacteria. Therefore enhance...” to “bacteria, therefore enhancing...” .

Response 47: “enhance” has been replaced by “enhancing” in line 247.

Point 48: In the discussion section, a reference should be cited. Xu X, Wei Y, Hua HW, Jing XQ, Zhu HL, Xiao K, Zhao JC, Liu YL*. 2022. Polyphenols sourced from Ilex latifolia Thunb. relieve intestinal injury via modulating ferroptosis in weanling piglets under oxidative stress. Antioxidants. 11:966.

Response 48: This reference has been added as reference [23] both in line 198 and 329-330.

Reviewer 2 Report

Comments to metabolites-2089468

In this study, Clostridium butyricum was used innovatively combined with Saccharomyces cerevisiae, and they played a crucial role in relieving the adverse effects of heat stress in goats. Probiotics have an effect on preventing heat stress, and this study added some novelty to the current study of applying probiotics. However, some details need to be modified and perfected. The point should be revised as follow:

In the full text: Change NH3-N to NH3-N” and “p < 0.05” to “p < 0.05”

Line 17-18: Change symbol ; to ,

Line 20: Delete the first and

Line 23: Change growth performance" to "growth performance of goats

Line 24: Change “the proper proportion of mixed probiotics” to exact supplementation levels

Line 41: Change [3. 4] to [3, 4]

Line 99: Delete concentrations

Line 102: Change CHOL (cholesterol) to cholesterol (CHOL)

Line 127: Delete were performed

Line 130: Change do to did

Line 131: Change Rectal to rectal

Line 154: Add that after than

Line 182: Delete Dietary supplementation with

Line 215: Change 3) to 4)

Line 223: Change same result to similar results

Line 225: Change toto from

Line 227: enhanceto enhancing

Author Response

Response to Reviewer 2 Comments

Dear reviewer, Thanks for your careful review and pertinent suggestions. As shown in the manuscript, we have made detailed modifications to the manuscript. Below you will find our point-by-point responses to your comments/ questions.

Point 1: In the full text: Change “NH3-N” to “NH3-N” and “p < 0.05” to “p < 0.05”

Response 1: “NH3-N” has been changed to “NH3-N” and “p < 0.05” has been
changed to “p < 0.05” in the full text.

Point 2: Line 17-18: Change symbol “;” to “,”

Response 2: “;” has been changed to “,” on lines 17-18.

Point 3: Line 20: Delete the first “and”

Response 3: the first “and” has been deleted at line 20.

Point 4: Line 23: Change “growth performance” to “growth performance of goats”

Response 4: “growth performance” has been changed to “growth performance of goats” in line 24.

Point 5: Line 24: Change “the proper proportion of mixed probiotics” to exact
supplementation levels

Response 5: “the proper proportion of mixed probiotics” has been replaced by “the mixture of 0.60% Saccharomyces cerevisiae and 0.05% Clostridium butyricum of the dry matter concentration” in lines 24-27.

Point 6: Line 41: Change “ [3. 4]” to “ [3, 4]”

Response 6: “[3. 4]” has been changed to “[3, 4]” at line 44.

Point 7: Line 99: Delete “concentrations”

Response 7: “concentrations” have been deleted at line 111.

Point 8: Line 102: Change “CHOL (cholesterol)” to “cholesterol (CHOL)”

Response 8: “CHOL (cholesterol)” has been changed to “cholesterol (CHOL)” in line 115. Point 9: Line 127: Delete “were performed” Response 9: “were performed” has been deleted at line 144.

Point 10: Line 130: Change “do” to “did”

Response 10: “do” has been changed to “did” at line 146.

Point 11: Line 131: Change “Rectal” to “rectal”

Response 11: “Rectal” has been changed to “rectal” in line 147.

Point 12: Line 154: Add “that” after “than”

Response 12: “that” has been added at line 172.

Point 13: Line 182: Delete “Dietary supplementation with”

Response 13: “Dietary supplementation with” has been deleted at line 203.

Point 14: Line 215: Change “3)” to “4)”

Response 14: “3)” has been changed to “ 4)” in line 238.

Point 15: Line 223: Change “same result” to “similar results”

Response 15: “same result” has been replaced by “similar results” at line 246.

Point 16: Line 225: Change “to”to “from”

Response 16: “to” has been changed to “from” in line 249.

Point 17: Line 227: “enhance” to “enhancing”

Response 17: “enhance” has been replaced by “enhancing” at line 250.

Reviewer 3 Report

General points about the manuscript: The manuscript brings interesting information regarding the feeding of different mixtures between Saccharomyces cerevisiae and Clostridium butyricum to goats. Authors evaluated physiological, blood biochemical, rumen fermentation and ruminal B vitamins parameters, as well as growth performance of the goats. The manuscript is very well written, with sufficient information in the different sections. I kindly pointed some issues that can improve the quality of this manuscript, please see below.

Specific considerations:

L10: Should you include abbreviation for M.L.?

L17-18: Replace semicolon by comma.

L17-19: This sentence in incomplete. It says the change (parameters that increased), but it does not say according to what. Is it a linear increase according to the treatments? Please rewrite it.

L21: Is it from P1 to P4? Or P1 and P4? Please correct it accordingly.

L24: Instead of saying “proper proportion”, please indicate more precisely what is the treatments (proportion between Saccharomyces cerevisiae and Clostridium butyricum) that gave the most beneficial results in this study.

Keywords: For indexing reasons, do not use as keywords those words already mentioned in the title. Choose as keywords, different words that those already cited in the title.

L30: Instead of restricting only to Jianghuai region in China, I would kindly suggest a broader approach, because goats are so often raised in hot areas all over the world. Thus, this study can be a reality of many places, not only Jianghuai region.

L33-35: This sentence should be reformulated. The idea is clear, but the English is broken. I would kindly suggest something like: “Our previous studies reported that goats kept in a high temperature environment for a period of time have significantly decreased rumen pH, ammonia-N and volatile fatty acids (VFAs) concentrations.”

L41: In the reference, should it be comma, instead of dot?

L42: This sentence is a bit lost, especially considering that readers might not know that Saccharomyces cerevisiae is a type of yeast. Please include a consideration connecting these ideas (Saccharomyces cerevisiae - yeast).

L46: Reference number 9 was mentioned, however references 7 and 8 were not mentioned yet. Please check and correct them mentioning in a numerical order. Please check references throughout the manuscript according to the Journal’s guideline.

Regarding the scientific names (Saccharomyces cerevisiae and Clostridium butyricum), include abbreviations first time mentioned in Introduction and then use only abbreviation (SC and CB) thereafter.

In the end of Introduction, clearly state the objective of this study.

L64: Precisely indicate the year, instead of saying “last year”.

L67: Please indicate if the amount (kg) of basal diet was provided in a fresh matter or dry matter.

Table 1: Please check the ingredient contents. The sum is a bit over 1000, maybe it’s just some rounding number. But just make sure the sum is 1000.

Table 1: Title – delete “The”, start with “Composition and nutri…”, it would sound better.

L73: Delete extra spaces.

L75: Thirty goats were divided

L83: Do you mean “periods”, instead of “cycles”? In my opinion, “period” sounds better when it concerns to Latin square design. If you agree, please check throughout.

L87: If so, mention that this is an adaptation period. And then, I believe during these 7 days, animals already received the mixture of Saccharomyces cerevisiae and Clostridium butyricum of each experimental treatment for adaptation. If so, mention this as well.

I would kindly suggest the authors to include a detailed description of the sampling during the experimental period. For instance, each period had 14 days, including 7 days for adaptation, and then, what has happened in day 8, day 9 and so on?

L98: Is the abbreviation superoxide dismutase? Spell out first time mentioned, and then only abbreviation. Check throughout the manuscript for every abbreviation. MDA as well in the next line.

L101: Is there a reason for Creatine, Urea and Sodium be with capital letter? Otherwise please correct.

L102: For cholesterol, indicate in the other way around. First spell out and then abbreviation between parentheses.

L103: For the signs (plus and minus) put them superscript.

Table 2: The information of this Table can be given in the text. I suggest writing the methods in the text along with each parameter just previously mentioned and then delete the Table.

L110: In NH3-N, the number 3 should be subscript.

L110: VFA was already previously described as abbreviation, then use only abbreviation. Please check throughout for all abbreviations used in this manuscript.

L112: Delete the space before the comma.

L115: For the reference between parenthesis, please be consistent. In other places it included also the state name, but not here. Just always mention in the same way.

L117: Do you mean “Dry matter intake”? Please check.

L118: Instead of the year, use the reference number according to Journal’s guideline.

L118: After period, start the new sentence with capital letter “The contents…”

L118: Abbreviation for “dry matter” could have been defined earlier, when shown for the first time.

L119: as described by

L120: The references mentioned here [21 and 22] does not match the reference list. Please check throughout the manuscript that the reference in text perfectly matches the reference in the list.

L121: Do you mean “NDF” instead of “DNF”?

L124: Be consistent with the reference between parentheses. Here it’s mentioned city, state, country. But this is not a pattern in the manuscript. Make this is a pattern. Also the state here was mentioned as abbreviation, but in other places it was spelled out (e.g., Massachusetts). Fix throughout the manuscript.

L131: Is there a reason to use capital R here? Other please change.

L132: It’s missing the second part of the parentheses.

Table 3: These physiological parameters described in Table 3 were not described in methodology. Please include them in Material and Methods.

Table 3 title: Replace “with” by “fed”. The same for the other titles.

L143: Insert space after “group”.

L143-144: Observing the results in the Table, this is not true when we check the differences statistically, as all of them have the superscript b. Please indicate differences only if they are statistically different or at least mention that numerically they can be different (but not statistically).

L146: Sentence is lacking full stop (dot).

L148: “different superscripts differ differently” does not sound good. Please rearrange this sentence. And fix for the other Tables as well.

L150: The number 3 should be subscript.

L150: Abbreviation TVFA was not defined. Check and correct for this abbreviation, as well as for all other abbreviations used.

L151: Explain the A/P ratio for example in methodology part, as well as in the Table 5. For example, in the Table, if you include “Acetic (A) acid” and “Propionic (P) acid”, then the reader will understand that A/P ratio is acetic divided by propionic acids.

L153-154: Observing the results in the Table, this is not true when we check the differences statistically, as all of them have the superscript b. Please indicate differences only if they are statistically different or at least mention that numerically they can be different (but not statistically).

Table 5: The enzymes described in this Table are not mentioned in methodology. Please include in Material and Methods.

L160: Use “…increased in the supplemented groups…” instead of “…increased in the P1-P4 groups…” Fix everywhere else.

Table 6: Include the unit in the title and then delete the row from the Table.

L166: The indication (a, b or a-b) is different in different Tables. Standardize them throughout.

L169-171: Observing the results in the Table, this is not true when we check the differences statistically, as all of them have the superscript b. Please indicate differences only if they are statistically different or at least mention that numerically they can be different (but not statistically).

L178 and L182: The sentence started in L178 and did not continue. Then the sentence started again in L182. Please correct it.

L181: Any reason to use capital B in Blue? Otherwise correct it.

L181: “indicates” instead of “indicated”.

L187: “…because previously we showed…”

L187: Include the reference number of your previous study. Is it reference 11?

L189: This paragraph and the next are aligned to the left. Please use the settings “justified” for both of them, as well as it was already done in the rest of the text.

L214-215: The reason number 3 is mentioned twice. Should you mention the last as number 4?

L216: “…can B-vitamins…” requires revision.

L218: Missing italic.

L225: “…could contribute to these probiotics could…” requires revision.

L229-231: This sentence was already described in the last sentence of the previous paragraph. Please do not repeat.

L230: “Dry matter” was previously defined as abbreviation, so please use abbreviation. Check throughout.

Can you include a brief consideration about the costs of using these probiotics in the diet of the goats? We see that the probiotics bring different advantages, but what about costs? Is it economically feasible? Does it pay back? What would be the additional cost per goat per day due to the probiotics supplementation?

L233: “supplementation”

L242: Delete semicolon after L. C.

L243: Delete semicolon after M. L.

L244: Who is X. Q. among the authors? Or did you mean Q. X.?

L252: You can mention that data can be obtained upon request, or something like that.

L263: “milk production” is misspelled.

L284: Indicate the scientific name in italic. Delete extra spaces.

L304: Provide full reference.

L309: Indicate the scientific name in italic.

L312: pH.

L319: China Agricultural University with capital letters.

L320: Instead of webpage, mention the DOI.

L322: Is this reference in Chinese? Was then the title here translated to English? Or is this available also in English. Please check.

L322: Should “Chin.” also be in italic? The first letters in the title are in capital letters, but not in other references, please check and include the references according to Journal’s guideline.

L323: Delete extra space before x.

L332: Delete extra space.

L335: Indicate the scientific name in italic.

L338: I could not find this reference, not even using the DOI. Please check if this is correct and also check that all other references are properly inserted, as well as DOI link working fine.

L340: Insert space before “Progress”.

L341: The year is not fully in bold.

Best regards.

Author Response

Response to Reviewer Comments

Dear reviewer, 

Thanks for your careful review and pertinent suggestions. As shown in the manuscript, we have made detailed modifications to the manuscript. Below you will find our point-by-point responses to your comments/ questions.

General points about the manuscript: The manuscript brings interesting information regarding the feeding of different mixtures between Saccharomyces cerevisiae and Clostridium butyricum to goats. Authors evaluated physiological, blood biochemical, rumen fermentation and ruminal B vitamins parameters, as well as growth performance of the goats. The manuscript is very well written, with sufficient information in the different sections. I kindly pointed some issues that can improve the quality of this manuscript, please see below.

Specific considerations:

Point 1: L10: Should you include abbreviation for M.L.?

Response 1: Thank you for your reminder. The abbreviation for M. L. has been added in line 10.

Point 2: L17-18: Replace semicolon by comma.

Response 2: Comma has been changed to a semicolon in lines 17-18.

Point 3: L17-19: This sentence in incomplete. It says the change (parameters that increased), but it does not say according to what. Is it a linear increase according to the treatments? Please rewrite it.

Response 3: The original sentence has been rewritten as your suggestion in lines 17-20.

Point 4: L21: Is it from P1 to P4? Or P1 and P4? Please correct it accordingly.

Response 4: P1 to P4 means P1, P2, P3, and P4.

Point 5: L24: Instead of saying “proper proportion”, please indicate more precisely what is the treatments (proportion between Saccharomyces cerevisiae and Clostridium butyricum) that gave the most beneficial results in this study.

Response 5: “the proper proportion of mixed probiotics” has been replaced by “the mixture of 0.60% Saccharomyces cerevisiae and 0.05% Clostridium butyricum of the dry matter concentration” in lines 24-26.

Point 6: Keywords: For indexing reasons, do not use as keywords those words already mentioned in the title. Choose as keywords, different words that those already cited in the title.

Response 6: Saccharomyces cerevisiae and Clostridium butyricum in the keywords section has been changed to probiotics in line 28.

Point 7: L30: Instead of restricting only to Jianghuai region in China, I would kindly suggest a broader approach, because goats are so often raised in hot areas all over the world. Thus, this study can be a reality of many places, not only Jianghuai region.

Response 7: Our studies are mainly carried out in the Jianghuai region. After reading this manuscript, we hope readers (researchers) can carry out relevant research in their own area.

Point 8: L33-35: This sentence should be reformulated. The idea is clear, but the English is broken. I would kindly suggest something like: “Our previous studies reported that goats kept in a high temperature environment for a period of time have significantly decreased rumen pH, ammonia-N and volatile fatty acids (VFAs) concentrations.”

Response 8: The original sentence has been rewritten in line with your suggestion in lines 34-37.

Point 9: L41: In the reference, should it be comma, instead of dot?

Response 9: [3.4] has been changed to [3, 4] in line 44.

Point 10: L42: This sentence is a bit lost, especially considering that readers might not know that Saccharomyces cerevisiae is a type of yeast. Please include a consideration connecting these ideas (Saccharomyces cerevisiae - yeast).

Response 10: As your suggestion, we made a note for Saccharomyces cerevisiae (a type of yeast) in line 42.

Point 11: L46: Reference number 9 was mentioned, however references 7 and 8 were not mentioned yet. Please check and correct them mentioning in a numerical order. Please check references throughout the manuscript according to the Journal’s guideline.

Response 11: References have been checked throughout the manuscript.

Point 12: Regarding the scientific names (Saccharomyces cerevisiae and Clostridium butyricum), include abbreviations first time mentioned in Introduction and then use only abbreviation (SC and CB) thereafter.

Response 12: To give readers a better reading experience, we think the full name is better.

Point 13: In the end of Introduction, clearly state the objective of this study.

Response 13: The objective of this study has been added in lines 57-59.

Point 14: L64: Precisely indicate the year, instead of saying “last year”.

Response 14: “last year” has been replaced by “2021” in line 66 .

Point 15: L67: Please indicate if the amount (kg) of basal diet was provided in a fresh matter or dry matter.

Response 15: Goats were fed on a basal diet in dry matter of 1.30 kg/day (line 69).

Point 16: Table 1: Please check the ingredient contents. The sum is a bit over 1000, maybe it’s just some rounding number. But just make sure the sum is 1000.

Response 16: I’m sorry we mistakenly put ground corn 261 to 267. In the revised manuscript, 267 has been replaced with 261 in Table 1.

Point 17: Table 1: Title – delete “The”, start with “Composition and nutri…”, it would sound better.

Response 17: Thank you for the recommendation, “ The composition and...”  has been changed to “Composition and...” in line 74.

Point 18: L73: Delete extra spaces.

Response 18: The extra spaces have been deleted in line 76.

Point 19: L75: Thirty goats were divided

Response 19: “Thirty goats have divided into...” has been changed to “Thirty goats were divided...” in line 78.

Point 20: L83: Do you mean “periods”, instead of “cycles”? In my opinion, “period” sounds better when it concerns to Latin square design. If you agree, please check throughout.

Response 20: In my opinion, “cycles” sounds better than “periods”.

Point 21: L87: If so, mention that this is an adaptation period. And then, I believe during these 7 days, animals already received the mixture of Saccharomyces cerevisiae and Clostridium butyricum of each experimental treatment for adaptation. If so, mention this as well.

Response 21: There is no adaptation period in this study. These 5 × 5 Latin square experiments were last from May to August (through the whole summer), which was close to the production practice of goats.

Point 22: I would kindly suggest the authors to include a detailed description of the sampling during the experimental period. For instance, each period had 14 days, including 7 days for adaptation, and then, what has happened in day 8, day 9 and so on?

Response 22: Thank you for your suggestion. Samples are only taken on each experimental cycle's last day (day 14). It has been described clearly in the original text, so we will not repeat it.  

Point 23: L98: Is the abbreviation superoxide dismutase? Spell out first time mentioned, and then only abbreviation. Check throughout the manuscript for every abbreviation. MDA as well in the next line.

Response 23: The full name of SOD, GSH-Px, and MDA have been added in lines 110-111.

Point 24: L101: Is there a reason for Creatine, Urea and Sodium be with capital letter? Otherwise please correct.

Response 24: The first letter of these three substances is not capitalized, and we have corrected it in the text in lines 114-115.

Point 25: L102: For cholesterol, indicate in the other way around. First spell out and then abbreviation between parentheses.

Response 25: “CHOL (cholesterol)” has been changed to “cholesterol (CHOL)” in line 115.

Point 26: L103: For the signs (plus and minus) put them superscript.

Response 26: As you suggested, we modified similar problems in the paper.

Point 27: Table 2: The information of this Table can be given in the text. I suggest writing the methods in the text along with each parameter just previously mentioned and then delete the Table.

Response 27: Tabulating the methods makes it easy for readers to read and reference these methods. So we decided to keep this table.

Point 28: L110: In NH3-N, the number 3 should be subscript.

Response 28: According to your suggestion, “NH3-N” has been changed to “NH3-N” throughout the manuscript.

Point 29: L110: VFA was already previously described as abbreviation, then use only abbreviation. Please check throughout for all abbreviations used in this manuscript.

Response 29: The full name of VFA has been deleted in line 123.

Point 30: L112: Delete the space before the comma.

Response 30: The space has been deleted at line 122.

Point 31: L115: For the reference between parenthesis, please be consistent. In other places it included also the state name, but not here. Just always mention in the same way.

Response 31: In parentheses, the name of the state or city has been added in lines 104-106, 117, 129, 130, and 140.

Point 32: L117: Do you mean “Dry matter intake”? Please check.

Response 32: Yes! Daily dry matter intake has been added in line 133.

Point 33: L118: Instead of the year, use the reference number according to Journal’s guideline.

Response 33: “Cai et al. [2021]” has been replaced by “Cai et al. [11]” in line 134.

Point 34: L118: After period, start the new sentence with capital letter “The contents…”

Response 34: “the” has been changed to “The...” in line 131. 

Point 35: L118: Abbreviation for “dry matter” could have been defined earlier, when shown for the first time.

Response 35: dry matter (DM), neutral detergent fiber (NDF), and acid detergent fiber (ADF) were not the first time they appear here. Therefore their abbreviation was used here in line 134.

Point 36: L119: as described by

Response 36: I’m sorry! What do you mean?

Point 37: L120: The references mentioned here [21 and 22] does not match the reference list. Please check throughout the manuscript that the reference in text perfectly matches the reference in the list.

Response 37: The references [21, 22] have been changed to [19, 20] after double checking in line 136.

Point 38: L121: Do you mean “NDF” instead of “DNF”?

Response 38: “DNF” has been changed to “NDF” in line 136.

Point 39: L124: Be consistent with the reference between parentheses. Here it’s mentioned city, state, country. But this is not a pattern in the manuscript. Make this is a pattern. Also the state here was mentioned as abbreviation, but in other places it was spelled out (e.g., Massachusetts). Fix throughout the manuscript.

Response 39: “(v4.0.5, GitHub Inc., San Francisco, CA, USA)” has been changed to “(v4.0.5) (GitHub Inc., San Francisco, USA)” in line 140.

Point 40: L131: Is there a reason to use capital R here? Other please change.

Response 40: “Rectal” has been changed to “rectal” in line 144.

Point 41: L132: It’s missing the second part of the parentheses.

Response 41: “(P1-P4 groups” has been deleted in line 146.

Point 42: Table 3: These physiological parameters described in Table 3 were not described in methodology. Please include them in Material and Methods.

Response 42: The measurement methods of physiological parameters were listed in lines 102-107.

Point 43: Table 3 title: Replace “with” by “fed”. The same for the other titles.

Response 43: “with” has been changed to “fed” in the title of table 3 to table 7.

Point 44: L143: Insert space after “group”.

Response 44: A space has been inserted after “group” in lines 158.

Point 45: L143-144: Observing the results in the Table, this is not true when we check the differences statistically, as all of them have the superscript b. Please indicate differences only if they are statistically different or at least mention that numerically they can be different (but not statistically).

Response 45: We reanalyzed the original data, found some errors, and corrected them.

Point 46: L146: Sentence is lacking full stop (dot).

Response 46: A dot has been added after “Table 4” in line 161.

Point 47: L148: “different superscripts differ differently” does not sound good. Please rearrange this sentence. And fix for the other Tables as well.

Response 47: In all the Tables, “different superscripts differ differently” have been changed to “Different letter superscripts indicate significant differences (p < 0.05)  and the same letter superscripts indicate insignificant differences (p >0.05).

Point 48: L150: The number 3 should be subscript.

Response 48: “NH3-N” has been changed to “NH3-N” throughout the manuscript.

Point 49: L150: Abbreviation TVFA was not defined. Check and correct for this abbreviation, as well as for all other abbreviations used.

Response 49: The full name of TVFA has been added in line 166.

Point 50: L151: Explain the A/P ratio for example in methodology part, as well as in the Table 5. For example, in the Table, if you include “Acetic (A) acid” and “Propionic (P) acid”, then the reader will understand that A/P ratio is acetic divided by propionic acids.

Response 50: The calculation method of the A/P ratio has been listed in lines 124-126.

Point 51: L153-154: Observing the results in the Table, this is not true when we check the differences statistically, as all of them have the superscript b. Please indicate differences only if they are statistically different or at least mention that numerically they can be different (but not statistically).

Response 51: We reanalyzed the original data, found some errors, and corrected them.

Point 52: Table 5: The enzymes described in this Table are not mentioned in methodology. Please include in Material and Methods.

Response 52: The methods for measuring these enzymes have been added in lines 131-132.

Point 53: L160: Use “…increased in the supplemented groups…” instead of “…increased in the P1-P4 groups…” Fix everywhere else.

Response 53: According to your suggestion, “…increased in the P1-P4 groups…” has been replaced by “…increased in each probiotics supplemented groups…” throughout the “Results” section.

Point 54: Table 6: Include the unit in the title and then delete the row from the Table.

Response 54: I’m sorry! To be consistent with the other tables in the article, We didn't make any adjustments to Table 6.

Point 55: L166: The indication (a, b or a-b) is different in different Tables. Standardize them throughout.

Response 55: The note below tables 4-7 has been changed to “Different letter superscripts indicate significant differences (p < 0.05) and the same letter superscripts indicate insignificant differences (p >0.05).

Point 56: L169-171: Observing the results in the Table, this is not true when we check the differences statistically, as all of them have the superscript b. Please indicate differences only if they are statistically different or at least mention that numerically they can be different (but not statistically).

Response 56: We reanalyzed the original data, found some errors, and corrected them.

Point 57: L178 and L182: The sentence started in L178 and did not continue. Then

the sentence started again in L182. Please correct it.

Response 57: The redundantly “Dietary supplementation with” has been deleted.

Point 58: L181: Any reason to use capital B in Blue? Otherwise correct it.

Response 58: “Blue” has been changed to “blue” in line 203.

Point 59: L181: “indicates” instead of “indicated”.

Response 59: The note below tables 4-7 has been changed to “Different letter superscripts indicate significant differences (p < 0.05) and the same letter superscripts indicate insignificant differences (p >0.05)”.

Point 60: L187: “…because previously we showed…”

Response 60: “because our previous showed” has been replaced with “showing” at 211.

Point 61: L187: Include the reference number of your previous study. Is it reference 11?

Response 61: Yes!

Point 62: L189: This paragraph and the next are aligned to the left. Please use the settings “justified” for both of them, as well as it was already done in the rest of the text.

Response 62: The format of this paragraph has been adjusted according to your suggestion.

Point 63: L214-215: The reason number 3 is mentioned twice. Should you mention the last as number 4?

Response 63: “3)” has been changed to “4)” in lines 237.

Point 64: L216: “…can B-vitamins…” requires revision.

Response 64: “…can B-vitamins…” has been replaced by “can product B-vitamins” in line 240.

Point 65: L218: Missing italic.

Response 65: “Clostridium butyricum” has been changed to “Clostridium butyricum” in line 243.

Point 66:L225: “…could contribute to these probiotics could…” requires revision.

Response 66: “These promote effects could contribute to these probiotics could...” has been replaced by “These promoted effects may be contributed from these probiotics could...” in lines 249-250.

Point 67:L229-231: This sentence was already described in the last sentence of the previous paragraph. Please do not repeat.

Response 67: The original sentence has been changed to “. In this study, the combination levels of 0.60% Saccharomyces cerevisiae and 0.05% Clostridium butyricum of dry matter are optimal supplemental levels to promote rumen fermentation and improve growth performance of goats in hot summer.” in lines 252-254.

Point 68:L230: “Dry matter” was previously defined as abbreviation, so please use abbreviation. Check throughout.

Response 68: “Dry matter” has been replaced by “DM” in line 253.

Point 69:Can you include a brief consideration about the costs of using these probiotics in the diet of the goats? We see that the probiotics bring different advantages, but what about costs? Is it economically feasible? Does it pay back? What would be the additional cost per goat per day due to the probiotics supplementation?

Response 69: Both saccharomyces cerevisiae and Clostridium butyricum were ¥25.00/ 500 g. According to the optimal addition ratio in this study (the mixture of 0.60% Saccharomyces cerevisiae and 0.05% Clostridium butyricum of the dry matter), The amount of these two probiotics required by each goat per day was 7.89 g and 0.65 g, respectively. In other words, the cost of probiotics was 0.427 yuan per goat per day. The feeding period is 14 days, so the total need is 5.56 yuan per goat. The addition of probiotics is cost-effective because it increases feed returns and reduces veterinary costs. We have not conducted an evaluation in the current study, and we will attach importance to this aspect for the assessment in future studies.

Point 70:L233: “supplementation”

Response 70: The “Conclusion” section has been rewritten.

Point 71:L242: Delete semicolon after L. C.

Response 71: the semicolon after L. C has been removed in line 269.

Point 72:L243: Delete semicolon after M. L.

Response 72: The semicolon after M. L. has been removed in line 270.

Point 73:L244: Who is X. Q. among the authors? Or did you mean Q. X.?

Response 73: “X. Q.” has been replaced by” Q. X.” in line 271.

Point 74:L252: You can mention that data can be obtained upon request, or something like that.

Response 74: I’m sorry! It isn’t necessary to mention this issue.

Point 75:L263: “milk production” is misspelled.

Response 75: The spelling of “milk production” has been corrected in line 290.

Point 76: L284: Indicate the scientific name in italic. Delete extra spaces.

Response 76: “Saccharomyces cerevisiae” has been changed to “Saccharomyces cerevisiae” and the extra spaces have been deleted in line 310.

Point 77: L304: Provide full reference.

Response 77: The full reference has been listed.

Point 78: L309: Indicate the scientific name in italic.

Response 78: “Saccharomyces cerevisiae has been changed to “Saccharomyces cerevisiae “ in line 342.

Point 79:L312: pH.

Response 79: “ph has been changed to “pH “ in line 340.

Point 80:L319: China Agricultural University with capital letters.

Response 80: “agricultural” has been changed to “Agricultural” in line 345.

Point 81:L320: Instead of webpage, mention the DOI.

Response 81: The webpages have been changed to DOI throughout the “reference” section.

Point 82:L322: Is this reference in Chinese? Was then the title here translated to English? Or is this available also in English. Please check.

Response 82: This article has been published in Chinese, but the title and abstract were also given in English.

Point 83:L322: Should “Chin.” also be in italic? The first letters in the title are in capital letters, but not in other references, please check and include the references according to Journal’s guideline.

Response 83: “Chin” has been changed to “Chin” in line 355.

Point 84:L323: Delete extra space before x.

Response 84: the extra space has been deleted in line 351.

Point 85:L332: Delete extra space.

Response 85: the extra space has been deleted.

Point 86:L335: Indicate the scientific name in italic.

Response 86: “Saccharomyces cerevisiae has been changed to “Saccharomyces cerevisiae.

Point 87:L338: I could not find this reference, not even using the DOI. Please check if this is correct and also check that all other references are properly inserted, as well as DOI link working fine.

Response 87: This reference is in Chiese only. The doi should be DOI: 10.3969/j.issn.1002-2813.2012.02.005. 

Point 88:L340: Insert space before “Progress”.

Response 88: A space has been added before “Progress” in line 373.

Point 89:L341: The year is not fully in bold.

Response 89: The year is fully in bold in line 374.

Reviewer 4 Report

ID: metabolites-2089468

Title: The mixture of Saccharomyces cerevisiae and Clostridium butyricum

could promote rumen fermentation and improve the growth performance of goats

in hot summer

The purpose of this study was to investigate the effects of multiple mixing ratio pairs of Saccharomyces cerevisiae and Clostridium butyricum supplementation on blood chemical parameters, rumen fermentation and growth performance of goats in hot summer. Although the work is interesting, some of those concern points and some questions are shown below;

Title: 

- The title should clearly state which probiotic is a supplement or component of the feed formulation. For example: “The supplementation with the mixture of ... goats in hot summer”.

Abstract:

-L19: “...were significantly increased” Which group?

-L22: “...the P3 group had the most significant...” Which statistical value or parameter indicated P3 had the most significant effect or optimal level in this study? /Please clearly demonstrate.

Introduction:

- There are many ways to reduce heat and heat stress. Why not improve the housing environment used to raise animals? / The author should state the limitations of other ways. /Why did the authors choose to use probiotic supplementation as a major issue to modify or improve animal performance in the summer?

-Please state the objectives and hypothesis of this study.

Materials and Methods: 

- Please specify the study month and ambient temperature.

-L67: “Goats were fed on a basal diet of 1.30 kg/day” Is it fed 1.30 kg/head/d or ad libitum? 

- How many times a day does the animal receive feed?

-L76-80: “These groups were as follows: ... of the dry matter weight of the basal diet” Please indicate more that this is supplemental in the diet or is used as a component of the feed formula for more clear understanding to the reader.

- How did the author supplement S. cerevisiae and C. butyricum? /What form or type of S. cerevisiae and C. butyricum were used in the study?

-L101-104: “The alanine transaminase ... analyzer (Hitachi 7100, Japan)” Please add the citations of the combining DiaSys Diagnostic System and the procedure of an automatic biochemical analyzer.

- How to measure the ruminal activities of avicelase, CMCaes, cellobiase, and xylanase 

 /Please indicate as well.

-L118: “Cai et al. [2021]”?? Please check the intext reference style. 

-L118: “the contents of the dry matter...” >>> “The...” not “the...”

-L120: “AOAC” There is no in the reference list. /Please check

-L120: “Goering and Van Soest” There is no in the reference list. /Please check

-L121, 122: “nutrients content in feedstuff”>>It should be changed to “nutrient content intake”.

-L124: “R studio” There is no in the reference list. /Please check

-Please demonstrate the statistical model used.

Results:

-L130: “Probiotics do not affect ...in hot summer” >>> change to “The supplementation of probiotics on physiological parameters of goats in hot summer”

-L136: “Probiotics affected blood biochemistry” >>> change to “The supplementation of probiotics on blood biochemistry”

-L145, 146: “Blood biochemical parameters ....in Table 4” Move to first sentence of this paragraph.

-L141: “However, the activity and ... control group (p < 0.05).” remove / It's a redundant sentence.

-L143: “The ascensional range of these parameters in P3 was greater than that in other probiotic groups.”?? Which statistical value is indicative? / It is shown by the superscript (b) the same as another group, which means not significantly different from another probiotic group.

-L144: “The rangeability of ... groups.” Not required to state this sentence because there is no statistical difference.

-L149: “Probiotics improve rumen fermentation” >>> change to “The supplementation of probiotics on rumen fermentation”

-L150: “TVFA” Use the full word first, then the abbreviation.

-L151: “A/P ratio” Use the full word first also.

-L152: “..in P1-P4 group compared...” >>> “...in probiotic supplemented (P1, P2, P3, and P4) group compared...”

L153, 154: “Moreover, the ascensional range ... probiotic groups.” Which statistical value is indicative? / It is shown by the superscript (b) the same as another group, which means not significantly different from another probiotic group.

-L154, 155: “The parameters of rumen fermentation ... in Table 5.” Move to first sentence of this paragraph.

-L158 “Probiotics promoted ruminal B-vitamins production in hot summer” >>> change to “The supplementation of probiotics on ruminal B-vitamins production”

-L160: “..in P1-P4 group compared...” >>> “...in probiotic supplemented group compared...”

-L162, 163: “The concentrations of ruminal B ...in Table 6.” Move to first sentence of this paragraph.

-L167: “Probiotics improve growth performance of goats in hot summer” >>> change to “The supplementation of probiotics on growth performance”

-L169, 170: “Moreover, the ascensional range of these parameters...probiotic groups.” Which statistical value is indicative? / It is shown by the superscript (b) the same as another group, which means not significantly different from another probiotic group.

-L171, 172: “The parameters of the growth performance ... in Table 7.” Move to first sentence of this paragraph.

Discussion:

- There are no discussions on blood biochemical parameters.

- There are no discussions about biological mechanisms on ruminal pH, NH3-N, avicelase, CMCaes, cellobiase, and xylanase. 

-L182: “Dietary” >>> “dietary”

-L195: “probiotics are beneficial...bacteria” How do probiotics benefit the growth and reproduction of cellulose-digesting bacteria in the rumen? /Please explain more about biological mechanisms.

-L216, 217: “the P3 group showed the most significant improvement in rumen fermentation.” Is it biased? /In this study, the P3 shown by the superscript (b) the same as another group, which means not significantly different from another probiotic group. / It should mention that the P3 showed the most significant improvement in ruminal B-vitamin concentrations.

-L227, 228: “the P3 group showed the most significant improvement in the digestibilities”? In this study, the P3 shown by the superscript (b) the same as another group, which means not significantly different from another probiotic group.

Conclusions:

- What about the physiological parameters and blood biochemical parameters?

-L235: “...vitamin B1, B2, and niacin...” remove and rewrite as a new sentence by the authors should indicate the level of probiotic mixture that affects these parameters because they have the highest statistical value.

References:

- Please check all the list

All table:

- Please indicate the meaning of P1, P2, P3, and P4 in the footnote of the table as well.

- All abbreviations should include the full word or meaning of each abbreviation in the footnote of the table as well.

- The superscript for a, b, or c indicating a greater value or a lower value? / Some parameters used a to indicate the highest value, and some parameters used a to indicate the lowest value. /Please double-check and modify to same all.

Table 1:

- The sum of total ingredient content = 1006? /Please double-check.

Table 2:

- Please add the citation of each method.

Table 6:

- “a-b Indicates within a row...” >>> “a, b, c Indicates within a row...”

Author Response

Response to Reviewer Comments

Dear reviewer, 

Thanks for your careful review and pertinent suggestions. As shown in the manuscript, we have made detailed modifications to the manuscript. Below you will find our point-by-point responses to your comments/ questions.

Point 1: Title:

- The title should clearly state which probiotic is a supplement or component of the feed formulation. For example: “The supplementation with the mixture of ... goats in hot summer”.

Response 1: We decided to maintain the current title, which accurately stated the conclusions of the study.

Abstract:

Point 2: -L19: “...were significantly increased” Which group?

Response 2: “...compared that of control.” has been added in line 20.

Point 3: -L22: “...the P3 group had the most significant...” Which statistical value or parameter indicated P3 had the most significant effect or optimal level in this study? /Please clearly demonstrate.

Response 3: The results showed that the parameters of rumen fermentation, including the pH values, the activities of ruminal cellulolytic enzymes, and the concentrations of ammonia nitrogen, acetic acid, propionic acid, total volatile fatty acid, vitamins B1, B2, and niacin were significantly increased in each probiotics supplemented group compared that of control. Moreover, the parameters of growth performance, including dry matter intake, average daily gain, the digestibilities of dry matter, neutral detergent fiber, and acid detergent fiber were significantly increased in each probioctics supplemented group compared to the control group. These parameters above increased the most in the P3 group, which was identified as the optimal supplemented group (level). Besides, this is a logical definition based on other studies related to dietary supplementation with probiotics and our previous research experience.

Introduction:

Point 4: - There are many ways to reduce heat and heat stress. Why not improve the housing environment used to raise animals? / The author should state the limitations of other ways. /Why did the authors choose to use probiotic supplementation as a major issue to modify or improve animal performance in the summer?

Response 4: 1) Unlike intensive pig farming, intensive goat farming requires giving them a certain amount of outdoor space. The outdoor temperature is uncontrollable in summer. 2) We're trying to solve real problems in the research areas that we're good at.

Point 5: -Please state the objectives and hypothesis of this study.

Response 5: The aim of this study has been added lines 57-59.

Materials and Methods:

Point 6: - Please specify the study month and ambient temperature.

Response 6: The feeding experiments in this study conduct were last from June to August, and the temperature was raised from 28.0 ± 2.1°C and 33.2 ± 2.7°C.

The temperature in this area can be referred to two references:

  • Sun, D., Deng, B., Wei, C. X., Jia, L., 2008. Effects of urbanization process on the climate in Hefei city. Journal of Anhui Agricultural Sciences2008, 26: 11484–11487.
  • Cai, Y., Yu, J.K., Zhang, J., Qi, D S. The effects of slatted floors and manure scraper systems on the concentrations and emission rates of ammonia, methane and carbon dioxide in goat buildings. Small Ruminant Research 2015, 132: 103-110.

Point 7: -L67: “Goats were fed on a basal diet of 1.30 kg/day” Is it fed 1.30 kg/head/d or ad libitum?

Response 7: Is it fed 1.30 kg/head/d. 

Point 8: - How many times a day does the animal receive feed?

Response 8: “The goats were fed the same diet twice daily (6:00-8:00 and 17:00–19:00).” has been added in line 70.

Point 9: -L76-80: “These groups were as follows: ... of the dry matter weight of the basal diet” Please indicate more that this is supplemental in the diet or is used as a component of the feed formula for more clear understanding to the reader.

Response 9: The probiotics were supplemented in the diet. “ The probiotics were supplemented in the diet ...” was added in line 79.

Point 10: - How did the author supplement S. cerevisiae and C. butyricum? /What form or type of S. cerevisiae and C. butyricum were used in the study?

Response 10: 1) The probiotics were mixed with the feedstuff according to the set supplemented level. 2) There's no specific strain. It's a mixture of strains. In the follow-up study, we will refine the research to specific strains.

Point 11: -L101-104: “The alanine transaminase ... analyzer (Hitachi 7100, Japan)” Please add the citations of the combining DiaSys Diagnostic System and the procedure of an automatic biochemical analyzer.

Response 11: The cities have been added in the reference in the parentheses. The analytical instrument is operated with one click, and it is not necessary to describe the inner workings of the machine in detail.

Point 12: - How to measure the ruminal activities of avicelase, CMCaes, cellobiase, and xylanase

Response 12: The measuring methods have been added in lines 124-126.

 /Please indicate as well.

Point 13: -L118: “Cai et al. [2021]”?? Please check the in text reference style.

Response 13: “Cai et al. [2021]” has been changed to  “Cai et al. [11]” in line 11.

Point 14: -L118: “the contents of the dry matter...” >>> “The...” not “the...”

Response 14: “the...” has been replaced by “The...” in line 134.

Point 15: -L120: “AOAC” There is no in the reference list. /Please check

Response 15: The reference has been added in lines 135-326. 

Point 16: -L120: “Goering and Van Soest” There is no in the reference list. /Please check

Response 16: The reference has been added in lines 135-327. 

Point 17: -L121, 122: “nutrients content in feedstuff”>>It should be changed to “nutrient content intake”.

Response 17: The reference has been added in lines 135-327. 

Point 18: -L124: “R studio” There is no in the reference list. /Please check

Response 18: R studio is a commonly used data analysis software, and it is not necessary to list it in the references.

Point 19: -Please demonstrate the statistical model used.

Response 19: The data of the rumen fermentation and growth performance parameters in the control group and each probioctics supplemented group were analyzed using a one-way analysis of variance (ANOVA) test followed by a post hoc Dunn test for multiple pairwise-comparison.

Results:

Point 20: -L130: “Probiotics do not affect ...in hot summer” >>> change to “The supplementation of probiotics on physiological parameters of goats in hot summer”

Response 20: The title is a summary of the results of this paragraph, which we thought would be easier for the reader to read at first glance. So the original title was kept.

Point 21: -L136: “Probiotics affected blood biochemistry” >>> change to “The supplementation of probiotics on blood biochemistry”

Response 21: The title is a summary of the results of this paragraph, which we thought would be easier for the reader to read at first glance. So the original title was kept.

Point 22:- L145, 146: “Blood biochemical parameters ....in Table 4” Move to first sentence of this paragraph.

Response 22: To ensure the consistency of the text of the results section, “Blood biochemical parameters ....in Table 4” is still retained in the original position.

Point 23: -L141: “However, the activity and ... control group (p < 0.05).” remove / It's a redundant sentence.

Response 23: The redundant sentence has been deleted.

Point 24: -L143: “The ascensional range of these parameters in P3 was greater than that in other probiotic groups.”?? Which statistical value is indicative? / It is shown by the superscript (b) the same as another group, which means not significantly different from another probiotic group.

Response 24: 1) By simply comparing the number sizes. 2) We reanalyzed the original data, found some errors, and corrected them.

Point 25: -L144: “The rangeability of ... groups.” Not required to state this sentence because there is no statistical difference.

Response 25: This is an important statement when all probiotics supplementation groups are influential, and we need to know which group is most effective. To make it clear to the reader that we are looking for efficiency and a more effective supplementation level.

Point 26: -L149: “Probiotics improve rumen fermentation” >>> change to “The supplementation of probiotics on rumen fermentation”

Response 26: The title is a summary of the results of this paragraph, which we thought would be easier for the reader to read at first glance. So the original title was kept.

Point 27: -L150: “TVFA” Use the full word first, then the abbreviation.

Response 27: The full name of TVFA has been added in line 166.

Point 28: -L151: “A/P ratio” Use the full word first also.

Response 28: The calculation method of the A/P ratio has been listed. The full word was given first in lines 124-126.

Point 29: -L152: “..in P1-P4 group compared...” >>> “...in probiotic supplemented (P1, P2, P3, and P4) group compared...”

Response 29: “…increased in the P1-P4 groups…” has been replaced by “…increased in each probiotics supplemented groups…” throughout the “Results” section.

Point 30: L153, 154: “Moreover, the ascensional range ... probiotic groups.” Which statistical value is indicative? / It is shown by the superscript (b) the same as another group, which means not significantly different from another probiotic group.

Response 30: 1) By simply comparing the number sizes. 2) We reanalyzed the original data, found some errors, and corrected them.

Point 31: -L154, 155: “The parameters of rumen fermentation ... in Table 5.” Move to first sentence of this paragraph.

Response 31: To ensure the consistency of the text of the results section, “The parameters of rumen fermentation ... in Table 5.” is still retained in the original position.

Point 32: -L158 “Probiotics promoted ruminal B-vitamins production in hot summer” >>> change to “The supplementation of probiotics on ruminal B-vitamins production”

Response 31: The title is a summary of the results of this paragraph, which we thought would be easier for the reader to read at first glance. So the original title was kept.

Point 33: -L160: “..in P1-P4 group compared...” >>> “...in probiotic supplemented group compared...”

Response 33: “…increased in the P1-P4 groups…” has been replaced by “…increased in each probiotics supplemented groups…” throughout the “Results” section.

Point 34: -L162, 163: “The concentrations of ruminal B ...in Table 6.” Move to first sentence of this paragraph.

Response 34: To ensure the consistency of the text of the results section, “The concentrations of ruminal B ...in Table 6.” is still retained in the original position.

Point 35: -L167: “Probiotics improve growth performance of goats in hot summer” >>> change to “The supplementation of probiotics on growth performance”

Response 35: The title is a summary of the results of this paragraph, which we thought would be easier for the reader to read at first glance. So the original title was kept.

Point 36: -L169, 170: “Moreover, the ascensional range of these parameters...probiotic groups.” Which statistical value is indicative? / It is shown by the superscript (b) the same as another group, which means not significantly different from another probiotic group.

Response 36: 1) By simply comparing the number sizes. 2) We reanalyzed the original data, found some errors, and corrected them.

Point 37: -L171, 172: “The parameters of the growth performance ... in Table 7.” Move to first sentence of this paragraph.

Response 37: To ensure the consistency of the text of the results section, “The parameters of the growth performance ... in Table 7.” is still retained in the original position.

Discussion:

Point 38:- There are no discussions on blood biochemical parameters.

Point 39:- There are no discussions about biological mechanisms on ruminal pH, NH3-N, avicelase, CMCaes, cellobiase, and xylanase.

Response 38 and 39: Our previous studies* have discussed these parameters so many times. This study focused mainly on the effects of probiotics on the production of VFA and B vitamins in rumen fermentation and on growth performance. So we focus on these aspects of the discussion.

*Some articles with Liyuan Cai from College of Animal Science and Technology, Huazhong Agricultural University, Wuhan 430070, China as the first or corresponding author.

Point 40: -L182: “Dietary” >>> “dietary”

Response 40: The redundantly “Dietary supplementation with” has been deleted.

Point 41: -L195: “probiotics are beneficial...bacteria” How do probiotics benefit the growth and reproduction of cellulose-digesting bacteria in the rumen? /Please explain more about biological mechanisms.

Response 41: The explanation has been listed in lines 236 to 241.

Point 42: -L216, 217: “the P3 group showed the most significant improvement in rumen fermentation.” Is it biased? /In this study, the P3 shown by the superscript (b) the same as another group, which means not significantly different from another probiotic group. / It should mention that the P3 showed the most significant improvement in ruminal B-vitamin concentrations.

Response 42: 1) By simply comparing the number sizes, To make it clear to the reader that we are looking for efficiency and more effective supplementation level. 2) We reanalyzed the original data, found some errors, and corrected them. 3) In The “Discussion” section, we mainly focus to explains why probiotics promote the production of vitamin B, which has been proposed in the “Results” section and will not be detailed in the discussion part.

Point 43: -L227, 228: “the P3 group showed the most significant improvement in the digestibilities”? In this study, the P3 shown by the superscript (b) the same as another group, which means not significantly different from another probiotic group.

Response 43: This is an important statement when all probiotics supplementation groups are effective in improving the digestibilities, and we need to know which group is most effective in this aspect. To make it clear to the reader that we are not only looking for efficiency but also more effective supplementation level.

Conclusions:

Point 44: -What about the physiological parameters and blood biochemical parameters?

Response 44: The conclusion of the physiological parameters and blood biochemical parameters has been added in lines 256-260.

Point 45: -L235: “...vitamin B1, B2, and niacin...” remove and rewrite as a new sentence by the authors should indicate the level of probiotic mixture that affects these parameters because they have the highest statistical value.

Response 45: According to your suggestion The “Conclusion” section has been rewritten.

References:

Point 46:- Please check all the list

Response 46: According to the guidelines, all the references have been checked to ensure they are in the right format.

All table:

Point 47: - Please indicate the meaning of P1, P2, P3, and P4 in the footnote of the table as well.

Response 47: The definition of P1, P2, P3, and P4 have been clearly defined in the “Material and methods section. In order to ensure the simplicity of the article, it is no longer marked below the table.

Point 48:- All abbreviations should include the full word or meaning of each abbreviation in the footnote of the table as well.

Response 48: All the abbreviations and the full word were given in the “Material and methods section. In order to ensure the simplicity of the article, it is no longer marked below the table.

Point 49:- The superscript for a, b, or c indicating a greater value or a lower value? / Some parameters used a to indicate the highest value, and some parameters used a to indicate the lowest value. /Please double-check and modify to same all.

Response 49: Different letter superscripts indicate significant differences (p < 0.05), and the same letter superscripts indicate insignificant differences (p >0.05). The a, b, and c indicate differences, not trends.

Table 1:

Point 50:- The sum of total ingredient content = 1006? /Please double-check.

Response 50: I’m sorry we mistakenly put ground corn 261 to 267. In the revised manuscript, 267 has been replaced with 261 in Table 1.

Point 51:- Please add the citation of each method.

Response 51: The method of K+, Na+, and Cl- measurement has been added in Table 2.

Table 6: 

Point 52:- “a-b Indicates within a row...” >>> “a, b, c Indicates within a row...”

Response 52: In all the Tables, “different superscripts differ differently” have been changed to “Different letter superscripts indicate significant differences (p < 0.05) and the same letter superscripts indicate insignificant differences (p >0.05).

Round 2

Reviewer 4 Report

Dear Authors, 

I would want to thank you for taking my comment into consideration and incorporating it into the manuscript. The remarks were almost entirely revised, but there are still a few points that need to be clarified.

Point 38:- There are no discussions on blood biochemical parameters.

Point 39:- There are no discussions about biological mechanisms on ruminal pH, NH3-N, avicelase, CMCaes, cellobiase, and xylanase.

Response 38 and 39: Our previous studies* have discussed these parameters so many times. This study focused mainly on the effects of probiotics on the production of VFA and B vitamins in rumen fermentation and on growth performance. So we focus on these aspects of the discussion.

---> You are unable to refer to the earlier study because those observations are being studied in this work currently being presented. It indicates that you gave it your whole attention as well. Therefore, biological mechanisms need to be presented in order for this work to be able to stand on its own!

--->Table 2,4,5 and 7 etc….all tables should stand alone and all abbreviations must be defined in a footnote.

Author Response

Point 1:- There are no discussions on blood biochemical parameters.

Response 1: In this study, we only focused on the changes in these biochemical parameters and did not focus on the mechanism of these changes, so we will not discuss them. The discussion would be dull even if we had a discussion based on phenomena. In addition, this is not our primary focus, so there is no need to discuss it.

Point 2:- There are no discussions about biological mechanisms on ruminal pH, NH3-N, avicelase, CMCaes, cellobiase, and xylanase.

Response 2: Our previous studies have discussed these parameters so many times. These studies as follow:

  1. Xue, L. G., Wang, D., Zhang, F. Y.,Cai, L. Y. Prophylactic feeding of Clostridium butyricum and Saccharomyces cerevisiae were advantageous in resisting the adverse effects of heat stress on rumen fermentation and growth performance of goats. Animals. 2022, 12: 2455.
  2. Xue, L. G., Zhou, S. Y., Wang, D., Zhang, F. Y., Li, J. F., Cai L. Y. The low dose of Saccharomyces cerevisiaeis beneficial for rumen fermentation (both in vivo and in vitro) and the growth performance of heat-stressed goats. Microorganisms. 2022, 10:
  3. Cai, L. Y., Hartanto, R., Xu, Q. B., Zhang, J.,Qi, D. S*. Saccharomyces cerevisiaeand Clostridium butyricum could improve B-vitamin production in the rumen and growth performance of heat-stressed goats. Metabolites. 2022, 12, 766.
  4. 4.Cai, L. Y., Hartanto, R., Zhang, J., Qi, D. S. Clostridium Butyricumimproves rumen fermentation and growth performance of heat-stressed goats in vitro and in vivo. Animals. 2021, 11: 3261.
  5. Cai, L. Y., Yu, J. K., Hartanto, R., Qi, D. S. Dietary supplementation with Saccharomycescerevisiae, Clostridium butyricumand their combination ameliorate rumen fermentation and growth performance of heat-stressed goats. Animals. 2021, 11: 2116.

Point 3: >Table 2,4,5 and 7 etc….all tables should stand alone and all abbreviations must be defined in a footnote.

Response 3: The grouping information has been added as a footnote in Tables 3-7.

Round 3

Reviewer 4 Report

The​ authors​ did​ not​ reply​ to​ second​ review​ comment.

Again​ here! 

---> You are unable to refer to the earlier study because those observations are being studied in this work currently being presented. It indicates that you gave it your whole attention as well. Therefore, biological mechanisms need to be presented in order for this work to be able to stand on its own!

Author Response

Dear reviewer, 

Thanks again for your careful review and pertinent suggestions. As shown in the manuscript, we have made detailed modifications to the manuscript. Below you will find our point-by-point responses to your comments/ questions.

Point 1: You are unable to refer to the earlier study because those observations are being studied in this work currently being presented. It indicates that you gave it your whole attention as well. Therefore, biological mechanisms need to be presented in order for this work to be able to stand on its own!

Response 1: Based on your suggestions, we have added content to the discussion section in lines 222-240 and lines 279-282.

Round 4

Reviewer 4 Report

Thank you for considering my suggestion to revise the manuscript. This amended version is more beneficial to the reader and is therefore recommended for publishing. Congratulation!

Author Response

Thank you very much!